# Accurate Discrimination of Mold-Damaged *Citri Reticulatae Pericarpium* Using Partial Least-Squares Discriminant Analysis and Selected Wavelengths

**DOI:** 10.3390/foods13233856

**Published:** 2024-11-29

**Authors:** Huizhen Tan, Yang Liu, Hui Tang, Wei Fan, Liwen Jiang, Pao Li

**Affiliations:** 1Guangdong Provincial Key Laboratory of Utilization and Conservation of Food and Medicinal Resources in Northern Region, Shaoguan University, Shaoguan 512005, China; thuizhen888@163.com (H.T.); lokeytang@163.com (H.T.); hnndjlw@163.com (L.J.); 2College of Food Science and Technology, Hunan Agricultural University, Changsha 410128, China; fs.ly@hunau.edu.cn (Y.L.); weifan@hunau.edu.cn (W.F.)

**Keywords:** *Citri Reticulatae Pericarpium*, mold-damaged, near-infrared spectroscopy, partial least-squares discriminant analysis, randomization test

## Abstract

Unscrupulous merchants sell the mold-damaged *Citri Reticulatae Pericarpium* (CRP) after removing the mold. In this study, an accurate and non-destructive strategy was developed for the discrimination of mold-damaged CRPs using portable near-infrared (NIR) spectroscopy and chemometrics. The outer surface and inner surface spectra were obtained without destroying CRPs. The discrimination models were established using partial least squares-discriminant analysis (PLS-DA) and wavelength selection strategy was used to further improve the discrimination ability. The predictive ability of models was assessed using the test set and an independent test set obtained one month later. The results demonstrate that the models of the outer surface outperform those of the inner surface. With multiplicative scatter correction (MSC)-PLS-DA, 100% accuracies were obtained in test and independent test sets. Furthermore, the wavelength selection strategy simplified the models with 100% discrimination accuracy. In addition, the randomization test (RT)-PLS-DA model developed in this study combines both the benefits of high accuracy and robustness, which can be applied for the accurate discrimination of mold-damaged CRPs.

## 1. Introduction

*Citri Reticulatae Pericarpium* (CRP) is among the most frequently used medicine and food homology substances [1,2]. There is a folk proverb that goes “CRP, the older, the better”. This means that storing it for a longer period enhances the quality of CRP [3]. However, storage under inappropriate temperature and humidity conditions may lead to mold growth in CRP. Changes in functional compounds often occur in mold-damaged CRP, which may be harmful to human health. Unscrupulous merchants sell the mold-damaged CRP after removing the mold as an undamaged one to gain higher profits. It is hard to discriminate between mold-damaged CRP after removing the mold from an undamaged one.

Many detection technologies have been applied for the analysis of CRP. Spectrophotometry has been used for the analysis of total flavonoids in CRP, while electrochemical and chromatographic methods have been applied to determine the contents of specific flavonoids [4]. In addition, the analysis of volatile compounds in CRP is mainly carried out using gas chromatography-mass spectrometry (GC-MS) [5,6,7]. However, these methods take a lot of time and come at a high cost. In addition, destructive sample pretreatment is required in these methods, while it is hard to achieve the non-destructive and rapid analysis of CRP. To our knowledge, no study exists on the detection of mold-damaged CRPs.

As one of the famous green analytical technologies, near-infrared (NIR) spectroscopy has been applied for the analysis of edible and medicinal homologous functional food [8,9,10]. The simultaneous discrimination of storage life and origin of CRPs was achieved with hand-held NIR spectrometer and multivariate analysis methods [11]. Portable NIR spectroscopy in diffuse reflectance mode was applied for the discrimination of different-age CRPs [12]. Fourier transform NIR spectroscopy in diffuse reflectance mode, integrating sphere diffuse and data in combination, was applied for the nondestructive discrimination of different-age CRPs [13]. Flash gas chromatography e-nose and Fourier transform NIR combined with a deep learning algorithm was successfully applied to distinguish different-aged CRPs [14]. However, there is no report on the detection of mold-damaged CRPs with NIR spectroscopy.

In the NIR analysis, the discrimination models are established using pattern recognition techniques. Partial least squares-discriminant analysis (PLS-DA) is among the popular supervised pattern recognition and multivariate dimensionality-reduction methods [15,16,17]. Based on our prior knowledge, the PLS-DA method may provide greater accuracy in recognition compared to the clustering analysis. However, the number of samples is usually much less than the number of variables, which can easily lead to chance discriminations (models that by chance give a good classification of the two groups) [18]. Therefore, the permutation test has been developed to evaluate the test of discrimination models [19]. In addition, a series of wavelength selection methods, such as projected variable importance (VIP), competitive adaptive reweighted sampling (CARS), and Monte Carlo uninformative variable elimination (MCUVE), were used for the dimensionality reduction in the PLS-DA method [20,21,22]. A randomization test (RT) has been developed to achieve wavelength (variable) selection and improve the predictive ability of the models by reducing the bias introduced by the uninformative wavelength [23]. The predictive ability of the partial least-squares (PLS) model was greatly improved with the selected informative variables using the RT method. However, this effective method has not yet been applied to the PLS-DA algorithm.

This study developed an accurate and non-destructive discrimination strategy for mold-damaged CRPs with portable NIR spectroscopy and chemometrics. Mold-damaged CRPs were obtained by placing the samples in the environment with high humidity for one month, while the undamaged CRP samples were stored in a dry place and away from light at the same time. The mold was wiped off with clean gauze. The outer surface and inner surface spectra were obtained without destroying the CRPs. The discrimination models were established using the PLS-DA pattern recognition method. In order to avoid the bias introduced by the uninformative wavelength in PLS-DA, a wavelength selection strategy was used. A new model named RT-PLS-DA for wavelength selection–pattern recognition was developed, compared with two existing wavelength selection–pattern recognition methods (CARS-PLS-DA and MCUVE-PLS-DA). The models were assessed using the test set and an independent test set obtained one month later.

## 2. Materials and Methods

### 2.1. Samples

In this study, mold-damaged CRP samples were prepared under controlled laboratory conditions. An NIR spectrometer was used to collect the spectra of mold-damaged and undamaged CRPs, while the chemometric methods were employed to analyze the spectral data, aiming to obtain the accurate discrimination model of mold-damaged CRPs. Five-year-old CRP samples were purchased from Guangdong Fu Dong Hai Co., Ltd. (Zhanjiang, China). Mold-damaged CRPs were obtained by placing the samples in the environment with high humidity for one month. In addition, consistent with the preservation method of long-term stored CRPs, 5-year-old CRPs in this study were stored in a dry place and away from light. The mold-damaged and undamaged CRPs were authenticated by experts in National and Local Joint Engineering Laboratory for Comprehensive Utilization of Citrus Resources in Hunan Academy of Agricultural Sciences. The mold was wiped off with clean gauze. In total, 567 CRP samples, including 289 undamaged and 278 mold-damaged samples, were collected. Changes in functional compounds may occur in the mold-damaged CRPs, which may be harmful to human health. Figure 1 shows the outer surface and inner surface photos of mold-damaged CRP after removing the mold and the undamaged one. As shown in the figures, there is no visual difference between the two types of CRPs in the photos for both the outer surface and inner surface. It is hard to accurately identify the mold-damaged CRPs.

### 2.2. Spectra Measurement

CRP spectra (890–1720 nm) were collected using a portable grating NIR spectrometer (i-Spec Plus, Metrohm, Switzerland) with the integrating sphere diffuse reflection mode. The light source was a 20 W halogen lamp. CRP samples were placed on top of the light spot, and the spectra of outer surface and inner surface were collected. One spectrum was obtained with one CRP sample. Each spectrum was the average of three repeated measurements of each sample and was composed of 511 data points. Therefore, 567 CRP spectra, including 289 undamaged and 278 mold-damaged, were collected. In addition, different samples of 100 undamaged and 100 mold-damaged CRPs were collected from the same manufacturer one month later. The spectra were obtained and used as the independent test set.

### 2.3. Data Analysis

The NIR analysis was often affected by the interferences of noise, peak overlapping, and baseline drift, which may decrease the accuracy of qualitative and quantitative analysis results. Spectral pretreatment can remove unnecessary information from spectral data to a certain extent, thereby improving the performance of prediction models. The baseline drift interference can be eliminated by using detrend (DT) and de-bias correction (de-bias) methods. The scatter-correction methods, including standard normal variate (SNV) transformation, multiplicative scatter correction (MSC), and normalization, were designed to reduce the physical variability between samples due to scatter, and can also be used to adjust for baseline shifts between samples [24]. In addition, maximum and minimum normalization (Min–Max) methods can settle the problem of data dispersion degree. A series of spectral derivative algorithms, including first-order derivative (1st), second-order derivative (2nd), and continuous wavelet transform (CWT), were applied to remove the high level of background and extract the effective information. The spectral pretreatment of the 1st removes only the baseline, while the spectral pretreatment of the 2nd removes both the baseline and the linear trend. In this study, the interference in the spectra of CRPs were removed by using eight spectral pretreatment methods (de-bias, DT, 1st, 2nd, CWT, Min–Max, MSC, and SNV).

In this research, the non-destructive discrimination models for mold-damaged CRPs were, respectively, obtained by using PCA (an unsupervised pattern recognition method) and PLS-DA (a supervised pattern recognition method). The calibration and test sets were obtained at a ratio of 2:1 using the Kennard–Stone (KS) method. Monte Carlo cross validation (MCCV) with adjusted Wold’s R criterion [25] was used for the determination of latent variable number for each PLS-DA model. In order to avoid the bias introduced by the uninformative wavelength in PLS-DA, wavelength selection strategy was used. A new model named RT-PLS-DA for wavelength selection–pattern recognition was developed, compared with two existing wavelength selection–pattern recognition methods (CARS-PLS-DA and MCUVE-PLS-DA). The RT method is based on the statistical significance test according to the randomized distribution, which effectively reduces the bias. With the principle of “survival of the fittest” from Darwin’s evolutionary theory, the CARS method can establish a high-performance qualitative model by selecting adaptable variables. In the MCUVE method, a large number of models by randomly selecting calibration samples were established and evaluated with the stability parameters of the variables in the models. The informative variables that represent the characteristics of the system can be obtained with the wavelength selection strategy, and the models can be simplified. In addition, the discriminative ability of the PLS-DA method may be enhanced by eliminating the redundant variables. The developed models were assessed using the test set and an independent test set obtained one month later. In addition, the 200 permutation test was used to evaluate the test of discrimination models by using multivariate data analysis software (SIMCA14.1 software package) [18,19,26]. The models were compared through discrimination accuracy of merit.

## 3. Results

### 3.1. Spectral Features and Discrimination Results of PCA

Figure 2A and Figure 2B show the NIR spectra of the outer surface and inner surface, respectively, which showed similar variation trends. The spectra of mold-damaged and undamaged CRPs were represented with different colors. It is hard to discriminate between the mold-damaged CRP and the undamaged one with the original spectra for both outer surface and inner surface sets.

In this study, the interferences of noise and baseline drift can be found in the NIR spectra. The interferences in the spectra of CRPs were removed by using eight spectral pretreatment methods (de-bias, DT, 1st, 2nd, CWT, Min-Max, MSC, and SNV). Figure 2C and Figure 2D show the spectra with SNV pretreatment of the outer surface and inner surface, respectively. SNV pretreatment can eliminate the baseline drift interference to a certain extent and extract effective information from the complex spectra. There are absorption peaks at 1200 nm and 1430 nm which belong to the O-H first overtone band and C-H second overtone band, respectively [11]. In addition, there are small differences in intensity between outer surface and inner surface spectra in the wavelength region of 1200–1400 nm. There is also obvious noise interference above 1600 nm and below 1000 nm. It is still hard to discriminate between the mold-damaged CRP and the undamaged one, even with the spectral pretreatment for both outer surface and inner surface datasets.

Figure 2E and Figure 2F show the PCA results of the outer surface and inner surface, respectively, based on the original spectra. The PCA dots of mold-damaged and undamaged CRPs were represented with different colors. The first two scores (PC1 and PC2) were used. The figures show the combination of groups of undamaged and mold-damaged CRPs. Figure 2G and Figure 2H show the PCA results of the outer surface and inner surface, respectively, based on the SNV pretreatment. However, there is a serious overlap between the undamaged and mold-damaged CRP data points, even with all the eight pretreatment methods. The unsupervised PCA method cannot provide accurate identification of any class. In addition, due to the lack of prior information, the discrimination ability of the unsupervised pattern recognition method is limited.

The PLS-DA model can provide good insight into the causes of discrimination via weights and loadings. Figure 3A and Figure 3B show the variable importance in the projection (VIP) and loading plots for all mold-damaged and undamaged CRP data. The value of the VIP can evaluate the contribution of each signal in the spectrum to the construction of the regression model. As shown in Figure 3A, the trends were similar for the inner surface and outer surface. The VIP values are above 1 within a wavelength range from 1350 to 1490 nm. This indicates that information from these wavelengths contributes more to the model. The wavelengths may be assigned to the O-H first overtone band. In addition, the obvious noise at wavelengths lower than 950 nm and higher than 1650 nm may reduce the discrimination accuracy. As shown in Figure 3B, distinct peaks at 1450 nm can be found in both inner surface and outer surface data, which assigns to the O-H first overtone band. In addition, the peak belonging to the C-H second overtone band at 1200 nm can be found in outer surface data. The information at 1200 and 1450 nm may contribute more to the model for discrimination mold-damaged CRP. Figure 3C shows the variation in error rates of PLS-DA with the numbers of latent variable between 1 and 15. The figure clearly shows that the optimal latent variable numbers for the outer surface and inner surface are 7 and 5, respectively.

### 3.2. Discrimination Results of PLS-DA

Based on prior knowledge, the PLS-DA method may obtain higher recognition accuracy compared to the clustering analysis. Figure 4A and Figure 4B show the discrimination accuracies of the test set with pretreatment methods and PLS-DA for the outer surface and inner surface, respectively. For both the outer surface and inner surface sets, the discrimination accuracies of the test set with PLS-DA and the original spectra are 100%. Both sensitivity and specificity rates are 100%. In addition, the discrimination accuracies of the test set with PLS-DA and pretreatment methods are 100%, except for the outer surface-2nd model (99%).

PLS-DA may be affected by biases introduced by non-informative wavelengths, especially when the number of variables is greater than that of samples. Using an independent test set is a good way to assess the test of discrimination models. Therefore, the PLS-DA methods were assessed using the independent test set in this study. Since the CRPs in the independent test set were obtained one month later, the distribution of the independent test set may be different from those of the calibration and test sets, which may lead to worse results for the independent test set. Figure 4C and Figure 4D show the discrimination accuracies of the independent test set with pretreatment methods and PLS-DA for the outer surface and inner surface, respectively. The discrimination accuracies of the independent test set are worse than those of the test set. For the outer surface and independent test set, the discrimination accuracy for the mold-damaged CRPs is only 78%, while the whole discrimination accuracy is 89% with PLS-DA and the original spectra. For the inner surface and independent test set, the discrimination accuracy for the mold-damaged CRPs is only 90%, while the whole discrimination accuracy is 95% with PLS-DA and the original spectra. The best model of the inner surface is MSC-PLS-DA, and the whole discrimination accuracy is 99%. The best model of the outer surface is also MSC-PLS-DA, and the whole discrimination accuracy can be reached, 100%. However, the discrimination accuracy for the mold-damaged CRPs is only 75% in the outer surface-2nd-PLS-DA model, while the whole discrimination accuracy is 88%. The discrimination accuracy for the mold-damaged CRPs is only 64% in the inner surface-2nd-PLS-DA model, while the whole discrimination accuracy is 82%. Pretreatment method of 2nd may increase noise level, and poor discrimination accuracy may be obtained with the inappropriate pretreatment method.

The results of the outer surface-MSC-PLS-DA model for the test and independent test sets are shown in Figure 5A and Figure 5B, respectively. The solid line stands for the threshold value and the two dashed lines signify the class values of mold-damaged and undamaged CRPs. The figures show that mold-damaged and undamaged CRPs were, respectively, classified into their groups for both the test and independent test sets. The robustness of outer surface-MSC-PLS-DA model was also evaluated using the 200 permutation test. Figure 5C shows the test plots of the outer surface-MSC-PLS-DA model. All blue Q^2^ points and green R^2^ points are under the starting point on the right. The blue regression line of the Q^2^ point with the vertical axis (left) is below zero. The low value of Q^2^ intercept indicates the robustness of the developed model. In conclusion, portable NIR spectroscopy combined with wavelength selection strategy-PLS-DA could quickly distinguish mold-damaged CRPs.

### 3.3. Discrimination Results of Wavelength Selection-PLS-DA

In order to avoid the bias introduced by the uninformative wavelength in PLS-DA, a wavelength selection strategy was used for the dimensionality reduction. The stability and prediction ability of the PLS-DA model may be improved by extracting the characteristic wavelengths. In this study, a new model named RT-PLS-DA for wavelength selection-pattern recognition was developed, compared with two existing wavelength selection-pattern recognition methods (CARS-PLS-DA and MCUVE-PLS-DA). The developed models were assessed using the test and independent test sets. Figure 6 shows the discrimination accuracies of the test set with wavelength selection-PLS-DA methods, while Figure 7 shows the discrimination accuracies of the independent test set. As shown in the figures, the best model for the original spectra is inner surface-RT-PLS-DA, and the whole discrimination accuracy of the independent test set can reach 96%. The model is better than that without wavelength selection (95%). In addition, the 100% discrimination accuracy of the test and independent test sets can be obtained with the outer surface models of CARS-SNV-PLS-DA, CARS-MSC-PLS-DA, RT-SNV-PLS-DA, MCUVE-SNV-PLS-DA, and MCUVE-MSC-PLS-DA. Both sensitivity and specificity rates are 100%. The results demonstrate that the models of the outer surface outperform those of the inner surface. Furthermore, only the RT-PLS-DA method can achieve 100% discrimination of the test set for the original spectra and all of the pretreatment methods. The RT-PLS-DA method developed in this study combines the benefits of high accuracy and robustness.

### 3.4. Variable Filtering Results

Figure 8 shows the variable numbers of CARS-PLS-DA, RT-PLS-DA, and MCUVE-PLS-DA models. For the outer surface and inner surface sets, the MCUVE method has the highest number of variables, followed by RT, and finally CARS. In addition, the wavelength selection results are not satisfactory in outer surface-MCUVE-MSC-PLS-DA, outer surface-MCUVE-SNV-PLS-DA, inner surface-MCUVE-original spectra-PLS-DA, inner surface-MCUVE-de-bias-PLS-DA, inner surface-MCUVE-Min-Max-PLS-DA, and inner surface-MCUVE-CWT-PLS-DA models. Up to 500 variables from a total of 511 wavelengths were selected in these models.

Figure 9 shows the wavelengths selected in outer surface-CARS-SNV-PLS-DA, outer surface-CARS-MSC-PLS-DA, outer surface-RT-SNV-PLS-DA, outer surface-MCUVE-SNV-PLS-DA, and outer surface-MCUVE-MSC-PLS-DA models obtained in Section 3.3. As shown in the figure, the wavelength selection results are not satisfactory with the MCUVE method. In addition, the ranges of the wavelength selected by CARS and RT are relatively similar. The variables were mostly distributed in the wavelength ranges of 1140–1250 nm, 1300–1430 nm, 1460–1510 nm, and 1540–1720 nm, which assign to the second overtone band of C-H, the first overtone band of O-H, the first overtone band of N-H, and the first overtone band of C-H, respectively [11]. These variables may be associated with the absorption of volatile oils, flavonoids, and alkaloids in CRP.

## 4. Conclusions

In this study, an accurate and non-destructive discrimination strategy for mold-damaged CRPs was developed with portable NIR spectroscopy and chemometrics. The results demonstrate that the models of the outer surface are better than those of the inner surface. For the outer surface, 100% accuracies of the test and independent test sets can be obtained using MSC-PLS-DA. Furthermore, the wavelength selection strategy in PLS-DA has the following advantages: (1) informative variables that represent the characteristics of the system can be obtained and the models can be simplified; (2) the discrimination ability of the PLS-DA models may be improved; and (3) the robustness of the models can be improved. The 100% discrimination accuracy of the test and independent test sets can be obtained with RT-PLS-DA models. In addition, the RT-PLS-DA method developed in this study combines the benefits of high accuracy and robustness. This research demonstrates the potential and procedure of applying NIR spectroscopy and pattern recognition techniques to CRP quality control. Using portable instruments, inferior CRPs can be quickly and accurately identified. The consumer will be protected against the intake of poor-quality products. Future research will focus on collecting CRP samples from different sources and establishing more effective models. In addition, hyperspectral imaging technology and deep learning algorithms will provide new methods for the non-destructive and accurate discrimination of mold-damaged CRPs.

## Figures and Tables

**Figure 1 foods-13-03856-f001:**
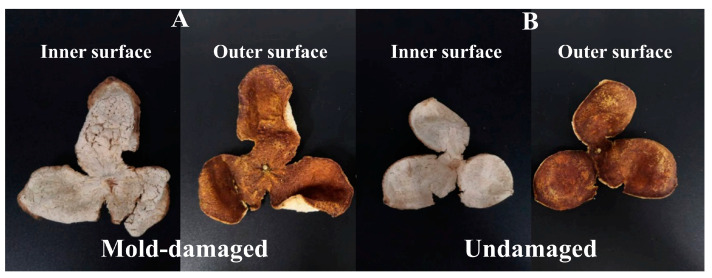
Inner surface and outer surface photos of mold-damaged CRP after removing the mold-damaged (**A**) and the undamaged one (**B**).

**Figure 2 foods-13-03856-f002:**
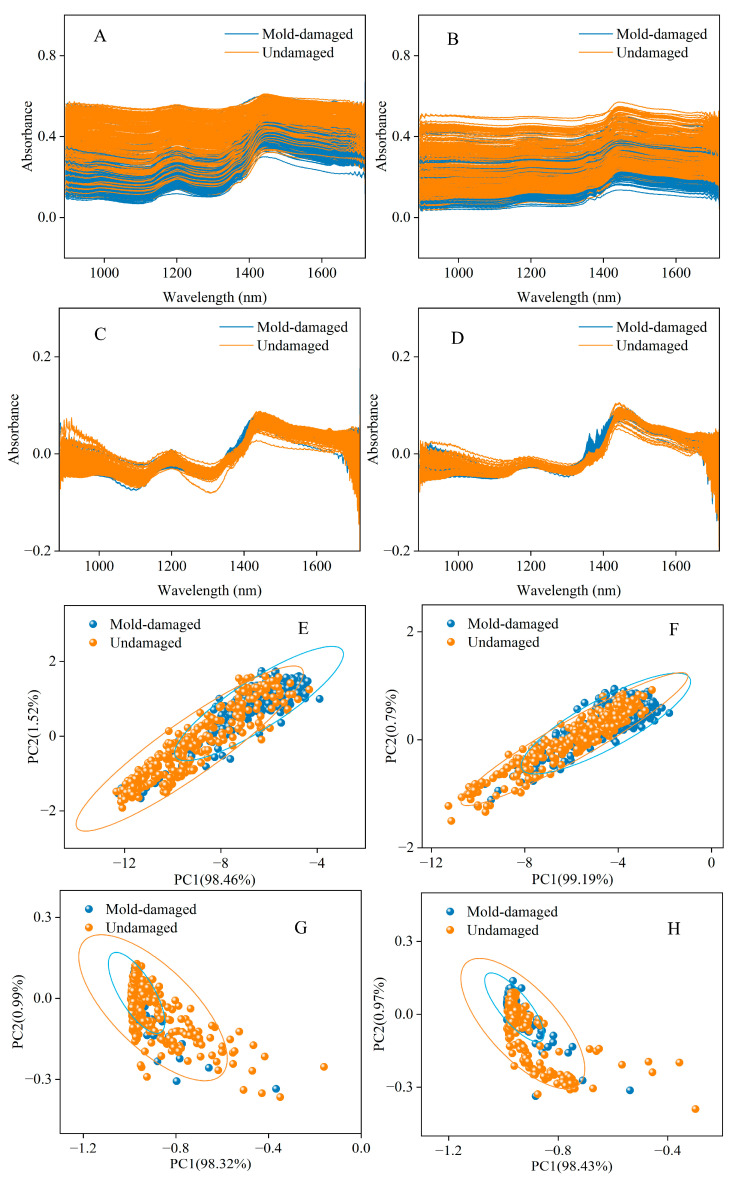
Original spectra of the outer surface (**A**) and inner surface (**B**). Spectra with SNV pretreatment of the outer surface (**C**) and inner surface (**D**). PCA results from the outer surface (**E**) and inner surface (**F**) with original spectra. PCA results from the outer surface (**G**) and inner surface (**H**) with the SNV method.

**Figure 3 foods-13-03856-f003:**
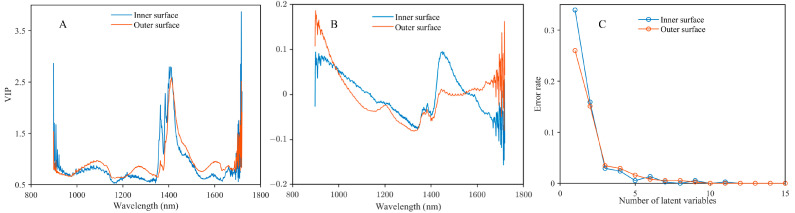
VIP (**A**) and loading (**B**) plots of PLS-DA model. Test of error rates of PLS-DA with the number of latent variables (**C**).

**Figure 4 foods-13-03856-f004:**
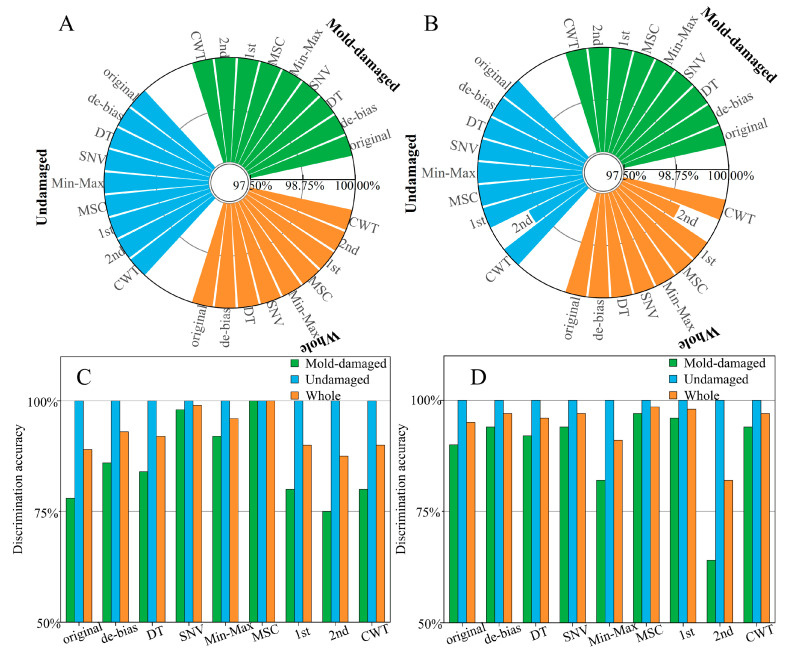
Discrimination accuracies of the test set with PLS-DA and pretreatment methods, for the outer surface (**A**) and inner surface (**B**). Discrimination accuracies of the independent test set with PLS-DA and pretreatment methods for the outer surface (**C**) and inner surface (**D**).

**Figure 5 foods-13-03856-f005:**
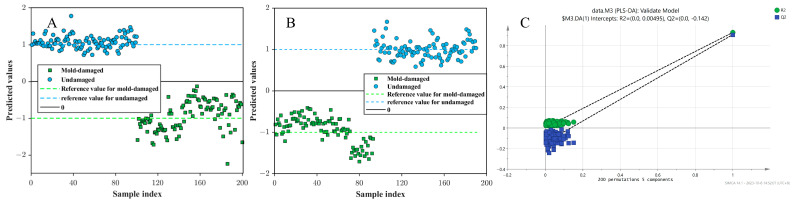
Results of outer surface-MSC-PLS-DA model for the test (**A**) and independent test sets (**B**). The test plots of outer surface-MSC-PLS-DA model (**C**).

**Figure 6 foods-13-03856-f006:**
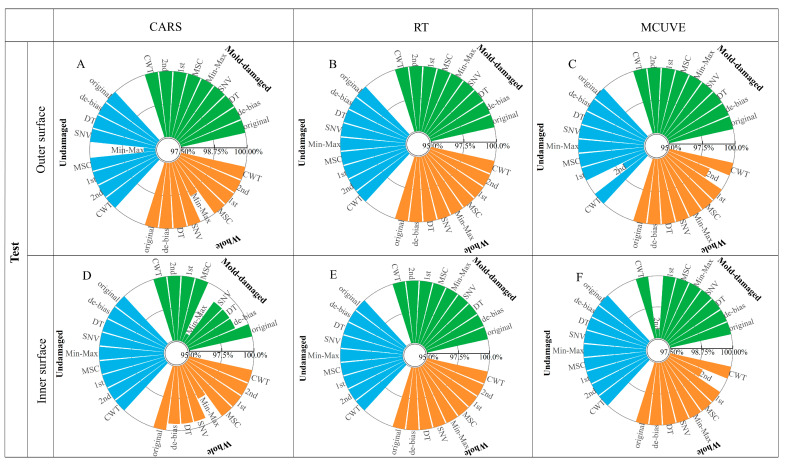
Discrimination accuracies of the test set with wavelength selection-PLS-DA methods (**A**–**F**).

**Figure 7 foods-13-03856-f007:**
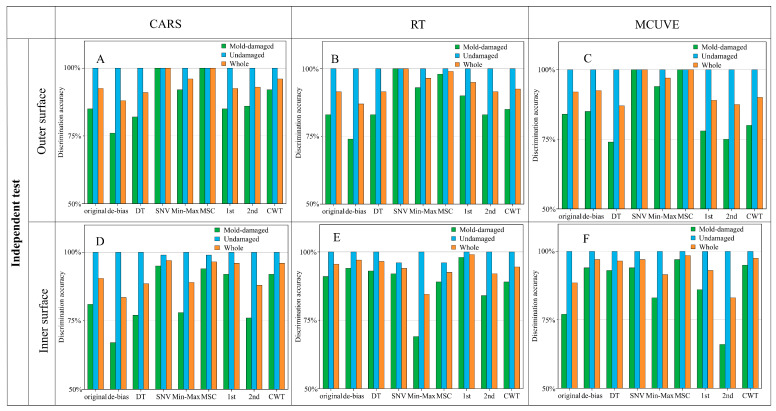
Discrimination accuracies of the independent test set with wavelength selection-PLS-DA methods (**A**–**F**).

**Figure 8 foods-13-03856-f008:**
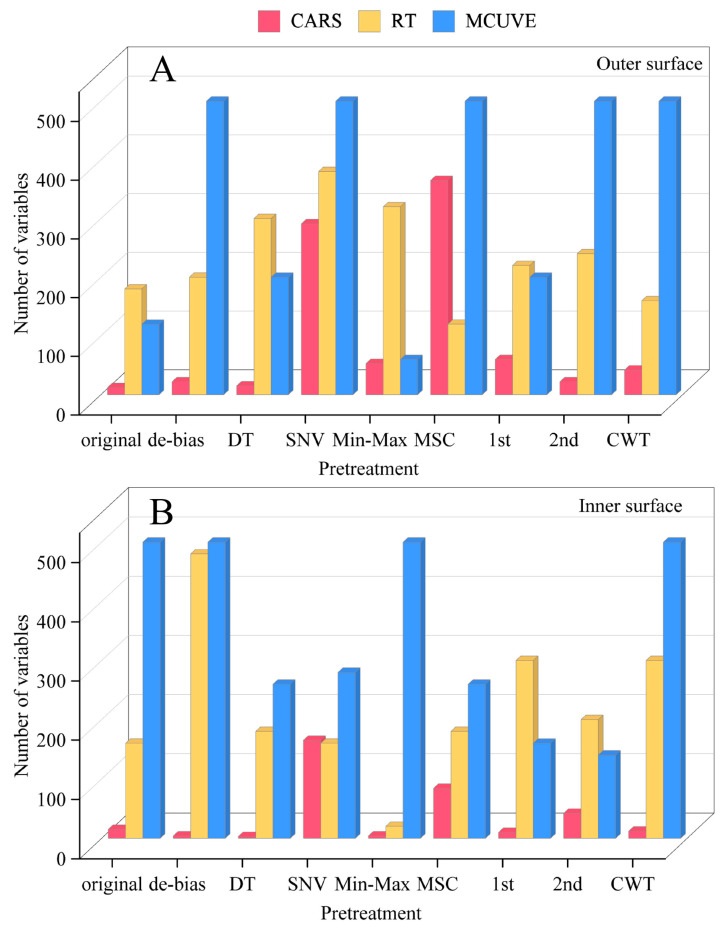
Variable numbers of CARS-PLS-DA, RT-PLS-DA, and MCUVE-PLS-DA models for the outer surface (**A**) and inner surface (**B**).

**Figure 9 foods-13-03856-f009:**
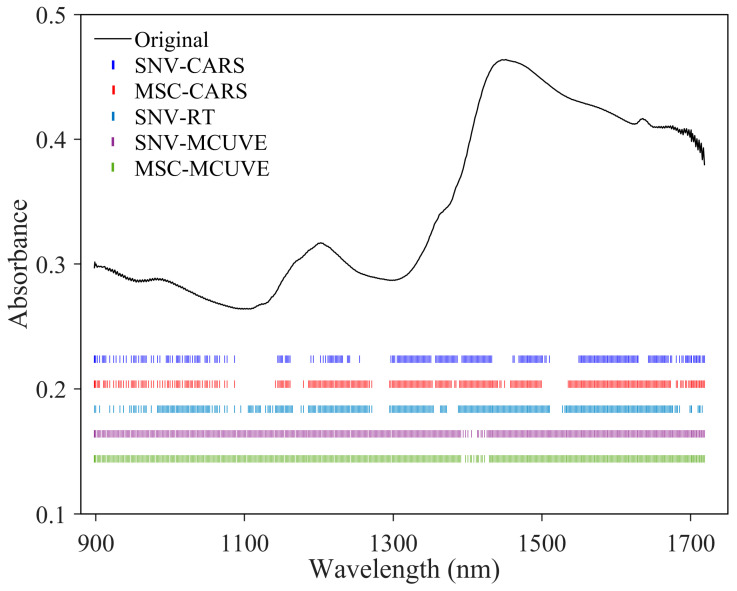
Wavelengths selected in outer surface-CARS-SNV-PLS-DA, outer surface-CARS-MSC-PLS-DA, outer surface-RT-SNV-PLS-DA, outer surface-MCUVE-SNV-PLS-DA, and outer surface-MCUVE-MSC-PLS-DA models.

## Data Availability

The original contributions presented in this study are included in the article. Further inquiries can be directed to the corresponding author.

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
