# Peer review of "Accurate Discrimination of Mold-Damaged Citri Reticulatae Pericarpium Using Partial Least-Squares Discriminant Analysis and Selected Wavelengths"

_foods, 2024, doi:10.3390/foods13233856_

Round 1
Reviewer 1 Report
Comments and Suggestions for Authors
This research was executed to develop discrimination models to classify infected and non-infected dry peels of citrus, even if “dodgy” merchants washed them. It is a very interesting topic in the industry, and the authors showed their chemometrics expertise by application of various pre-treatment techniques to the original spectra.
However, I have the comments below to further improve the delivery of findings:
Title: The title is too long and misguiding. It reads as if the spectrometer was added to the PLS-DA models. I suggest “Accurate discrimination of mold-damaged Citri Reticulatae Pericarpium using partial least square discriminant analysis from selected wavelengths” or any simplified title.
Abstract:
L21-L22: With multiplicative scatter correction (MSC)-PLS-DA, 100% accuracies can be obtained in test and independent test sets. Why is this hypothetical? What did you find that suggests this hypothesis?
L22-L23: Furthermore, wavelength selection strategy can simplify the models with 100% discrimination accuracies. Why is this hypothetical? What did you find that suggests this hypothesis?
L23-L24: Besides, randomization test (RT)-PLS-DA model developed in this study combines both the benefits of high accuracy and robustness. Here you can mention a conclusive statement like what is your finding contributing to research or technical practices of discriminating the peels.
Keywords: Mention words associated to your title, do not just repeat words from the title. Words that can also be used to search your document on the internet.
L16; L77; L105: instead of writing capsule (which differs in meaning, regionally), I suggest inner surface.
L38; L97: remove “for the layman”.
L108-L110: from the same samples? Otherwise, explain the source of the samples as explained for the calibration samples.
L314-315: “The results demonstrate that the models of outer surface are better than those of internal capsule”. On this, I challenge the authors to develop PLS-DA models using the range 1300–1500nm after applying SNV pre-treatment.
Conclusions:Only results are mentioned in this section. Conclusions should be based on recommendations, this study limitations, future studies, or the findings potential for catching the unscrupulous merchants.
Comments on the Quality of English LanguageI suggest English language editing. Some minor grammatical errors were noticed but my review is based on the better delivery of findings.
Reviewer 2 Report
Comments and Suggestions for Authors
Dear Authors, you should address my comments highlighted across the text.

The English could be improved to more clearly express the research. Particularly, some sentences could be joined to make reading more fluent.
Reviewer 3 Report
Comments and Suggestions for Authors
In my opinion, it is very useful to use non-destructive methods of determination in food samples, so the topic of paper ``Accurate and non-destructive discrimination of mold-damaged Citri Reticulatae Pericarpium by wavelength selection strategy-partial least squares-discriminant analysis combined with portable near-infrared spectroscopy`` by Huizhen Tan et al. is very interesting. Statistical studies are done in a logical order and the results are presented nicely. However, the present work should be little more improved before being considered for publication.
- Have you confirmed results from NIR analysis with some destructive and standard method that mold has appeared under those conditions (Lines 87-89)? Also, since you used 5 year old CRPs, are you sure that there is no mold yield in normal samples?
- In the Figure 1, mark with 1 what is the outer surface and with 2 what is the internal capsule (or vice versa). When someone looks without delving into the text to know which part of the cortex is which.
- The sentence conifrm with literature, Line 166, ``There are absorption peaks at 1200 nm and 1430 nm, which belong to the O-H first overtone band and C-H second overtone band, respectively``. Did you have any expectations about which peaks and what intensity would occur in normal and which in mold-damage samples?
- Does figure 2 refer to normals and mold-damage together or just one group of these two groups?
- For easier reading of the results, move Figure 4 before discussing Figure 5.
- Also the sentence confirm with literature, Line 302 `` The variables were mostly distributed in the wave length ranges of 1140-1250 nm, 1300-1430 nm, 1460-1510 nm, and 1540-1720 nm, which assign to the second overtone band of C-H, the first overtone band of O-H, the first overtone band of N-H, and the first overtone band of C-H, respectively``.
Round 2
Reviewer 1 Report
Comments and Suggestions for Authors
Comments addressed adequately